# Association between sleep duration and hypertension in southwest China: a population-based cross-sectional study

Xiaoyu Chang ,[1] Xiaofang Chen,[2] John S Ji ,[3,4] Guojin Luo,[5] Xiaofang Chen*,[5] Qiang Sun,[5] Ningmei Zhang,[1] Yu Guo,[6] Pei Pei,[6] Liming Li,[7,8] Zhengming Chen,[9] Xianping Wu[10]

For numbered affiliations see end of article.

**Correspondence to**
Dr Xianping Wu;
wwwuxp@163.com

## ABSTRACT

**Objective** Hypertension is a major risk factor and cause of many non-communicable diseases in China. While there have been studies on various diet and lifestyle risk factors, we do not know whether sleep duration has an association to blood pressure in southwest China. This predictor is useful in low-resource rural settings. We examined the association between sleep duration and hypertension in southwest China.

**Design** Population-based cross-sectional study.

**Setting** This study was part of the baseline survey of a large ongoing prospective cohort study, the China Kadoorie Biobank. Participants were enrolled in 15 townships of Pengzhou city in Sichuan province during 2004–2008.

**Participants** 55 687 participants aged 30–79 years were included. Sleep duration was assessed by a self-reported questionnaire.

**Main outcome measures** Hypertension was defined as systolic blood pressure ≥140 mm Hg and/or diastolic blood pressure ≥90 mm Hg, or prior physician-diagnosed hypertension in hospitals at the township (community) level or above.

**Results** The prevalence of hypertension was 25.17%. The percentages of subjects with sleep durations of <6, 6, 7, 8 and ≥9 hours were 17.20%, 16.14%, 20.04%, 31.95% and 14.67%, respectively. In multivariable-adjusted analyses, the increased ORs of having hypertension were across those who reported ≥9 hours of sleep (men: 1.16, 95% CI 1.04 to 1.30; women: 1.19, 95% CI 1.08 to 1.32; general population: 1.17, 95% CI 1.08 to 1.26). The odds of hypertension was relatively flat until around 6.81 hours of sleep duration and then started to increase rapidly afterwards in subjects and a J-shaped pattern was observed. There was a U-shaped relationship between sleep duration and hypertension in females.

**Conclusion** Long sleep duration was significantly associated with hypertension and a J-shaped pattern was observed among rural adults in southwest China, independent of potential confounders. However, this association was not obvious between short sleep duration and hypertension.

## Strengths and limitations of this study

⇒ This study examined the association between sleep duration and hypertension in southwest China, and this predictor was useful in low-resource rural settings.

⇒ A large sample of rural population was obtained, which could represent most of the rural places in southwest China.

⇒ A broad range of covariates were controlled in the analysis, including age, gender, education, occupation, marital status, income, tea consumption, smoking, alcohol, metabolic equivalent, sedentary leisure time, fresh fruit consumption, insomnia, sleep snoring, body mass index, waist circumference, diabetes mellitus, stroke and coronary heart disease.

⇒ The dose–response relationship between sleep duration and hypertension was explored and visualised by restricted cubic spline regression.

⇒ The sleep duration was self-reported without more detailed and objective measures of sleep, which was less accurate than actigraphy and polysomnography.

cardiovascular disease (CVD) burden and mortality.[2] In China, hypertension is also a serious public health problem. The China hypertension survey showed that the prevalence of hypertension and prehypertension was 23.2% and 41.3%, respectively, among Chinese adults, which means that there is one patient with hypertension per four people.[3] High systolic blood pressure (SBP) was the leading risk factor contributing to deaths and disability-adjusted life years.[4] In addition, hypertension is a growing problem in urban and rural China. The prevalence of hypertension has sharply increased in rural China,[5] which is similar in rural and urban settings (23.4% vs 23.1%).[3] Disease burden of hypertension is increasing globally, especially as places of poverty become more prosperous.[2]

Previous studies about influencing factors of hypertension mainly focus on diet (high dietary sodium intake), exercise and other

## INTRODUCTION

Hypertension is a condition in which the blood vessels have persistently raised pressure.[1] It is the leading risk factor for global

lifestyle factors,[6 7] while some studies in recent years documented the relationship between sleep and hypertension. Sleep status influences blood pressure by altering autonomic nervous system function and other physiological events. Unhealthy sleep status such as sleep apnoea, insomnia, long and short sleep duration could increase the risk of hypertension. Even though a variety of studies examined sleep duration to be associated with hypertension, the results were not completely consistent. Some studies showed that short sleep duration increased the risk of hypertension,[8–14] others showed that long sleep duration was related to hypertension or the associations were J-shaped or U-shaped curves.[15–18] However, most of the study populations were from western countries. A small number of studies were conducted in Asian countries,[9 10 13 19] and there was limited information about the association between sleep duration and hypertension in southwest China. Therefore, the aim of this study is to analyse the baseline data from a large prospective cohort study of the China Kadoorie Biobank (CKB) to explore the association of sleep duration with hypertension among more than 50 000 adults in southwest China.

## SUBJECTS AND METHODS
### Study population
This study uses a large, ongoing prospective cohort study, the CKB. The baseline survey of CKB was conducted in five urban and five rural areas across China from 2004 to 2008. Our data were derived from the baseline survey of CKB in Sichuan province. A total of 55 687 residents aged 30–79 years in 15 townships of Pengzhou city in Sichuan province had completed baseline information collection, including a questionnaire, anthropometric measurements, blood sample collection and a written informed consent. More details about the study design are available in previous publications.[20–22]

### Data collection
The baseline survey took place in temporary study clinics, specially set up near participants' residential areas. Information on sociodemographic characteristics (age, gender, education level, occupation, marital status, annual household income), behavioural risk factors for chronic diseases (diet, tea consumption, smoking status, alcohol intake, physical activity, sedentary leisure time, sleep) and medical history was collected through an interviewer-administered computerised questionnaire by face-to-face interview with trained interviewers according to standard protocols and procedures. A range of physical measurements were also undertaken, including height, weight, waist circumference (WC) and so on.

### Outcome measurement ascertainment
Participants were asked to report the number of hours they slept a day during the last 12 months. Sleep duration was assessed by a self-reported questionnaire with the following question: 'On average, how many hours do you typically sleep per day (including daytime naps)?'. Respondents could report in only 1-hour increments. To maintain consistency with previous literatures,[10 17 18 23] sleep duration was categorised as five groups of <6, 6, 7 (which regarded as the reference group for our analysis), 8 and ≥9 hours.

As described in the previous study,[22] blood pressure was measured twice using a UA-779 digital monitor recommended by the British Hypertension Society (A & D Instruments, Abingdon, UK). If the SBP difference between the first two measurements was more than 10 mm Hg, a third measurement was obtained and the last two measurements were recorded. The mean of two blood pressure measurements was used for this study. Hypertension was defined as systolic blood pressure (SBP) ≥140 mm Hg and/or diastolic blood pressure (DBP) ≥90 mm Hg, or prior physician-diagnosed hypertension in hospitals at the township (community) level or above.

Several variables were used both as covariates in adjusted models and as variables for stratification. Sociodemographic characteristics included age at baseline survey (30–79 years old), gender (male or female), education level (no formal school, primary school, middle school, high school and above), occupation (farmer or others), marital status (married or never married/widowed/separated/divorced) and annual household income (<¥5000, ¥5000–¥9999, ≥¥10 000). Behavioural risk factors included tea consumption (never or almost never, only occasionally, usually at least once a week), smoking status (never, ex-regular, occasionally, daily or almost every day), alcohol intake (never regular, ex-regular, occasionally or seasonally, usually at least once a week), fresh fruit consumption (daily, 4–6 days/week, 1–3 days/week, monthly, never/rarely), physical activity, sedentary leisure time (hours/week), insomnia symptoms (yes or no), sleep snoring (never, occasionally, usually), WC and body mass index (BMI). Medical history included diabetes mellitus (DM), stroke and coronary heart disease (CHD). BMI was calculated as weight in kilograms (kg)/height in metres (m)$^2$. According to the Guidelines for Prevention and Control of Overweight and Obesity in Adults in China,[24] BMI was divided into four groups: underweight (BMI<18.5 kg/m$^2$), normal weight (18.5≤BMI<24 kg/m$^2$), overweight (24≤BMI<28 kg/m$^2$) and obesity (BMI≥28 kg/m$^2$). WC beyond 85 cm for men and 80 cm for women was recommended as central obesity.

Information about type, frequency and duration of occupational, commuting-related, household and active recreational (leisure-time) physical activities was used to calculate the total physical activity in metabolic equivalents (MET, hours/day).[25 26] Sedentary leisure time (hours/week) was assessed by the question: 'About how many hours per week did you watch TV or read?'. Insomnia symptoms were determined by three questions: taking >30 min to fall asleep after going to bed or waking up in the middle of the night; waking up early and not being able to go back to sleep; having difficulty staying alert while at work, eating or

meeting people during daytime. Those reporting 'Yes' for one or more of the three questions were classified as having insomnia symptoms.[27] Participants who consumed fresh fruits more than 3 days/week were defined as regular consumers, otherwise, those were never or occasionally consumers.[25]

## Statistical analysis

Continuous variables were expressed as mean and SD. Frequency and percentage distributions were reported for categorical variables. Continuous variables included age, MET, sedentary leisure time, BMI, WC, SBP and DBP. Categorical variables included age groups, gender, education level, occupation, marital status, income, tea consumption, smoking status, alcohol intake, fresh fruit consumption, insomnia, sleep snoring, BMI groups, DM, stroke and CHD. Prevalence of hypertension was calculated for the overall population and different subgroups, including sleep duration and some potential confounding factors. $\chi^2$ tests were conducted to test for group differences in prevalence or percentage, and the Cochran-Armitage tests for trend were used to examine the trends for ordered categorical variables. After examining the normality of the continuous covariates, the analysis of variance was used for continuous variables of study participants by different sleep durations. The statistical significance of associations for the continuous and categorical variables with sleep duration was estimated. Multivariable logistic regression models were used to calculate the ORs and 95% CIs for hypertension between sleep duration subgroups adjusted for potential confounders, including age, gender, education level, occupation, marital status, income, tea consumption, smoking status, alcohol intake, MET, sedentary leisure time, fresh fruit consumption, insomnia, sleep snoring, BMI, WC, DM, stroke and CHD. In addition, we performed subgroup analysis stratified by health-related factors, such as age, gender, tea consumption, smoking status, alcohol intake and insomnia. Potential non-linear association between sleep duration and hypertension was examined with restricted cubic spline regression. We used restricted cubic spline regression with five knots at the 5th, 35th, 50th, 65th and 95th centiles to flexibly model the association of sleep duration and hypertension.[28–30] Multiple linear regression model was used to explore the relationship for sleep duration with SBP and DBP. A two-tailed p value <0.05 was considered statistically significant. All data processing and statistical analyses were performed using Statistical Analysis Software V.9.4 (SAS Institute) and R V.4.0.3 (R Foundation for Statistical Computing, Vienna, Austria).

## Patient and public involvement

No patients were involved in the present study, the interpretation of study results or write-up of the manuscript. The results were presented to study participants on the

CKB website (http://www.ckbiobank.org/site/) and newsletters.

## RESULTS

### Different demographic characteristics of study population across sleep duration groups

The baseline characteristics of study population for subgroups according to sleep duration were presented in table 1. A total of 55 687 adults completed baseline information, including 34 372 women and 21 315 men. The mean (±SD) age of study population was 51.03±10.54 years (range: 30–79 years). The mean (±SD) sleep duration was 7.05±1.68 hours, which were 7.07±1.61 hours in men and 7.04±1.72 hours in women. The overall means (±SD) of SBP and DBP were 129.24±19.01 and 77.23±10.23 mm Hg, respectively. The percentages of the subjects with sleep durations of <6, 6, 7, 8 and ≥9 hours were 17.20%, 16.14%, 20.04%, 31.95% and 14.67%, respectively.

Individuals with less hours of sleep per day were more likely to be female, elderly, low education level, farmer, single, low annual household income, never or almost never tea consumers, low frequency of fresh fruit consumption and low BMI. Short sleepers were more likely to be DM, CHD, stroke or hypertensive subjects. With the increase in age, the proportion of subjects who slept for equal to or more than 8 hours decreased, while the proportion of subjects who slept for equal to or less than 6 hours increased. With the increase in education level, the proportion of subjects with less than 6 hours of sleep duration decreased. With the increase in annual household income and tea consumption, the proportion of subjects who slept for less than 6 hours decreased. With the increase in BMI and fresh fruit consumption, the proportion of subjects who slept for less than 6 hours decreased, while the proportion of subjects who slept for equal to or more than 9 hours increased.

### The prevalence of hypertension among adults in southwest China

The results of prevalence of hypertension in different gender groups among 55 687 adults were presented in table 2. 25.17% of 55 687 adults had hypertension. A quarter of participants were considered hypertensive, slightly more in men (28.67%) than in women (22.99%). The prevalence of hypertension in both men and women increased with age. The prevalence of hypertension in both genders decreased with increased education level, annual household income and tea consumption. Among different alcohol intake and smoking status groups, the highest prevalences of hypertension were observed in ex-regular drinker and ex-regular smoker in both males and females.

The prevalence of hypertension increased with decreased fresh fruit consumption in males and females. The prevalence of hypertension in both males and females increased with BMI and sleep snoring frequency. Higher prevalence of hypertension was also associated

**Table 1** Distributions of sleep duration by categorical basic characteristics and descriptive statistics of continuous covariates in the sleep duration groups

| Variables | Sleep duration (hours) | | | | | Overall | χ²/F | P value |
|---|---|---|---|---|---|---|---|---|
| | <6 | 6 | 7 | 8 | ≥9 | | | |
| Subjects, n (%) | 9576 (17.20) | 8988 (16.14) | 11 159 (20.04) | 17 790 (31.95) | 8174 (14.67) | 55 687 (100) | | – |
| Gender, n (%) | | | | | | | | |
| Female | 6271 (18.24) | 5273 (15.34) | 6594 (19.18) | 10 898 (31.71) | 5336 (15.53) | 34 372 (100) | 171.051 | <0.0001 |
| Male | 3305 (15.51) | 3715 (17.43) | 4565 (21.42) | 6892 (32.33) | 2838 (13.31) | 21 315 (100) | | |
| Age (years), n (%) | | | | | | | | |
| 30–39 | 604 (6.21) | 998 (10.27) | 1802 (18.54) | 4142 (42.60) | 2176 (22.38) | 9722 (100) | 4406.276 | <0.0001 |
| 40–49 | 1651 (11.01) | 2087 (13.91) | 3170 (21.13) | 5523 (36.82) | 2571 (17.13) | 15 002 (100) | | |
| 50–59 | 3560 (19.69) | 3278 (18.13) | 3815 (21.11) | 5262 (29.11) | 2161 (11.96) | 18 076 (100) | | |
| 60–69 | 2775 (27.41) | 2050 (20.25) | 1962 (19.38) | 2340 (23.12) | 996 (9.84) | 10 123 (100) | | |
| 70–79 | 986 (35.67) | 575 (20.80) | 410 (14.83) | 523 (18.92) | 270 (9.78) | 2764 (100) | | |
| Education level, n (%) | | | | | | | | |
| No formal school | 2610 (30.38) | 1555 (18.10) | 1452 (16.90) | 1942 (22.60) | 1033 (12.02) | 8592 (100) | 2440.546 | <0.0001 |
| Primary school | 5146 (18.53) | 4759 (17.14) | 5539 (19.95) | 8401 (30.26) | 3920 (14.12) | 27 765 (100) | | |
| Middle school | 1444 (9.70) | 2040 (13.70) | 3059 (20.55) | 5722 (38.44) | 2621 (17.61) | 14 886 (100) | | |
| High school and above | 376 (8.46) | 634 (14.27) | 1109 (24.95) | 1725 (38.82) | 600 (13.50) | 4444 (100) | | |
| Occupation, n (%) | | | | | | | | |
| Farmer | 8674 (17.73) | 7862 (16.07) | 9629 (19.68) | 15 531 (31.74) | 7236 (14.78) | 48 932 (100) | 102.183 | <0.0001 |
| Others | 902 (13.35) | 1126 (16.67) | 1530 (22.65) | 2259 (33.44) | 938 (13.89) | 6755 (100) | | |
| Marital status, n (%) | | | | | | | | |
| Married | 8040 (15.91) | 7990 (15.81) | 10 236 (20.26) | 16 610 (32.87) | 7650 (15.15) | 50 526 (100) | 797.729 | <0.0001 |
| Never married/widowed/separated/divorced | 1536 (29.76) | 998 (19.34) | 923 (17.88) | 1180 (22.86) | 524 (10.16) | 5161 (100) | | |
| Annual household income (¥), n (%) | | | | | | | | |
| <5000 | 3730 (24.61) | 2454 (16.19) | 2761 (18.22) | 3996 (26.37) | 2215 (14.61) | 15 156 (100) | 991.831 | <0.0001 |
| 5000–9999 | 3145 (15.98) | 3168 (16.09) | 3867 (19.65) | 6532 (33.18) | 2972 (15.10) | 19 684 (100) | | |
| ≥10 000 | 2701 (12.96) | 3366 (16.15) | 4531 (21.73) | 7262 (34.83) | 2987 (14.33) | 20 847 (100) | | |
| Tea consumption, n (%) | | | | | | | | |
| Never or almost never | 2311 (23.51) | 1534 (15.61) | 1768 (17.99) | 2692 (27.39) | 1525 (15.50) | 9830 (100) | 581.800 | <0.0001 |
| Only occasionally | 4822 (17.76) | 4488 (16.53) | 5385 (19.83) | 8695 (32.02) | 3764 (13.86) | 27 154 (100) | | |
| Usually at least once a week | 2443 (13.06) | 2966 (15.86) | 4006 (21.42) | 6403 (34.24) | 2885 (15.42) | 18 703 (100) | | |
| Smoking status, n (%) | | | | | | | | |
| Never | 4896 (16.38) | 4418 (14.78) | 5771 (19.30) | 9998 (33.44) | 4814 (16.10) | 29 897 (100) | 350.690 | <0.0001 |

Continued

**Table 1** Continued

| Variables | Sleep duration (hours) | | | | | Overall | $\chi^2/F$ | P value |
|---|---|---|---|---|---|---|---|---|
| | <6 | 6 | 7 | 8 | ≥9 | | | |
| Ex-regular smoker | 835 (22.14) | 675 (17.90) | 786 (20.84) | 996 (26.41) | 479 (12.71) | 3771 (100) | | |
| Occasional smoker | 892 (20.14) | 827 (18.67) | 892 (20.14) | 1272 (28.71) | 547 (12.34) | 4430 (100) | | |
| Daily or almost every day | 2953 (16.79) | 3068 (17.44) | 3710 (21.09) | 5524 (31.41) | 2334 (13.27) | 17 589 (100) | | |
| Alcohol intake, n (%) | | | | | | | 204.099 | <0.0001 |
| Never regular drinker | 3591 (18.80) | 2926 (15.32) | 3575 (18.72) | 6031 (31.56) | 2979 (15.60) | 19 102 (100) | | |
| Ex-regular drinker | 463 (22.43) | 371 (17.97) | 439 (21.27) | 512 (24.81) | 279 (13.52) | 2064 (100) | | |
| Occasional or seasonal drinker | 3548 (16.40) | 3456 (15.97) | 4439 (20.51) | 7118 (32.89) | 3078 (14.23) | 21 639 (100) | | |
| Usually at least once a week | 1974 (15.32) | 2235 (17.35) | 2706 (21.01) | 4129 (32.05) | 1838 (14.27) | 12 882 (100) | | |
| Fresh fruit consumption, n (%) | | | | | | | 1185.806 | <0.0001 |
| Daily | 736 (11.53) | 863 (13.52) | 1376 (21.56) | 2218 (34.76) | 1188 (18.63) | 6381 (100) | | |
| 4–6 days/week | 716 (12.75) | 814 (14.50) | 1134 (20.20) | 1931 (34.39) | 1020 (18.16) | 5615 (100) | | |
| 1–3 days/week | 3688 (15.35) | 3720 (15.49) | 4654 (19.38) | 8347 (34.75) | 3611 (15.03) | 24 020 (100) | | |
| Monthly | 3401 (21.93) | 2890 (18.63) | 3210 (20.69) | 4262 (27.48) | 1748 (11.27) | 15 511 (100) | | |
| Never/rarely | 1035 (24.88) | 701 (16.85) | 785 (18.87) | 1032 (24.81) | 607 (14.59) | 4160 (100) | | |
| BMI (kg/m²), n (%) | | | | | | | 161.222 | <0.0001 |
| <18.5 | 625 (25.31) | 436 (17.66) | 462 (18.71) | 642 (26.00) | 304 (12.32) | 2469 (100) | | |
| 18.5–23.9 | 5523 (17.33) | 5094 (15.99) | 6399 (20.08) | 10 209 (32.04) | 4642 (14.56) | 31 867 (100) | | |
| 24–27.9 | 2700 (16.12) | 2718 (16.23) | 3393 (20.26) | 5416 (32.34) | 2519 (15.05) | 16 746 (100) | | |
| ≥28 | 728 (15.81) | 740 (16.07) | 905 (19.65) | 1523 (33.07) | 709 (15.40) | 4605 (100) | | |
| DM, n (%) | | | | | | | 118.858 | <0.0001 |
| No | 9050 (16.89) | 8606 (16.06) | 10 751 (20.07) | 17 257 (32.21) | 7908 (14.77) | 53 572 (100) | | |
| Yes | 526 (24.87) | 382 (18.06) | 408 (19.29) | 533 (25.20) | 266 (12.58) | 2115 (100) | | |
| CHD, n (%) | | | | | | | 37.361 | <0.0001 |
| No | 9501 (17.15) | 8929 (16.11) | 11 101 (20.03) | 17 734 (32.00) | 8147 (14.71) | 55 412 (100) | | |
| Yes | 75 (27.27) | 59 (21.45) | 58 (21.09) | 56 (20.36) | 27 (9.83) | 275 (100) | | |
| Stroke, n (%) | | | | | | | 23.279 | 0.0001 |
| No | 9509 (17.17) | 8941 (16.14) | 11 115 (20.06) | 17 719 (31.99) | 8112 (14.64) | 55 396 (100) | | |
| Yes | 67 (23.02) | 47 (16.15) | 44 (15.12) | 71 (24.40) | 62 (21.31) | 291 (100) | | |
| Hypertension, n (%) | | | | | | | 362.709 | <0.0001 |
| No | 6530 (15.67) | 6527 (15.66) | 8473 (20.33) | 13 865 (33.27) | 6278 (15.07) | 41 673 (100) | | |
| Yes | 3046 (21.74) | 2461 (17.56) | 2686 (19.17) | 3925 (28.01) | 1896 (13.52) | 14 014 (100) | | |

**Table 1** Continued

| Variables | Sleep duration (hours) | | | | | Overall | $\chi^2$/F | P value |
|---|---|---|---|---|---|---|---|---|
| | <6 | 6 | 7 | 8 | ≥9 | | | |
| Age (years), mean (SD) | 56.30±9.94 | 53.37±10.19 | 50.80±10.07 | 48.60±10.09 | 47.90±10.31 | 51.03±10.54 | 1224.030 | <0.0001 |
| MET (hours/day), mean (SD) | 21.08±11.47 | 21.98±11.89 | 22.67±12.03 | 22.72±11.65 | 21.12±11.70 | 22.07±11.76 | 51.170 | <0.0001 |
| WC (cm), mean (SD) | 77.45±9.27 | 78.27±9.11 | 78.16±8.98 | 78.07±8.96 | 77.99±9.05 | 78.00±9.06 | 11.800 | <0.0001 |
| BMI (kg/m$^2$), mean (SD) | 23.00±3.29 | 23.29±3.21 | 23.32±3.18 | 23.38±3.15 | 23.39±3.19 | 23.29±3.20 | 25.120 | <0.0001 |
| Sedentary leisure time (hours/week), mean (SD) | 25.15±10.22 | 25.83±10.45 | 25.98±10.50 | 26.85±10.53 | 27.28±11.39 | 26.28±10.61 | 64.490 | <0.0001 |
| SBP (mm Hg), mean (SD) | 132.28±20.24 | 130.23±19.23 | 128.61±18.73 | 127.81±18.18 | 128.52±18.96 | 129.24±19.01 | 99.140 | <0.0001 |
| DBP (mm Hg), mean (SD) | 77.65±10.51 | 77.34±10.34 | 76.96±10.24 | 77.02±9.97 | 77.45±10.28 | 77.23±10.23 | 8.940 | <0.0001 |

The percentages in the table are row percent. $\chi^2$ tests were conducted to test for differences in percentage for categorical variables. One-way analysis of variance (ANOVA) tests were conducted to test for differences in means for categorical variables.
BMI, body mass index; CHD, coronary heart disease; DM, diabetes mellitus; MET, metabolic equivalent; SBP, systolic blood pressure; SD, standard deviation; WC, waist circumference.

with insomnia, DM, stroke and CHD in both males and females. The prevalence of hypertension in patients with DM (48.98%) was twice as high as that in non-DM subjects (24.23%). The prevalence of hypertension in patients who had a stroke (73.54%) was nearly triple of that in non-stroke subjects (24.91%). The prevalence of hypertension in both males and females decreased with sleep duration.

The results of prevalence of hypertension and distribution of patients with hypertension in different sleep duration groups were presented in table 3. In different sleep groups, the prevalences of hypertension were more in men than in women. The prevalence of hypertension increased with age. The prevalence of hypertension decreased with increased fresh fruit consumption. Among different alcohol intake and smoking status groups, the highest prevalences of hypertension were observed in ex-regular drinkers and ex-regular smokers in different sleep duration groups. Among 7 and 8 hours' sleepers, the prevalence of hypertension was higher among insomnia subjects than that among non-insomnia groups.

The percentages of the patients with hypertension with sleep durations of <6, 6, 7, 8 and ≥9 hours were 21.74%, 17.56%, 19.17%, 28.00% and 13.53%, respectively. Patients with less hours of sleep were more likely to be female, elderly, never or almost never tea consumers and low frequency of fresh fruit consumption. With the increase in age, the proportion of patients with hypertension who slept for equal to or more than 8 hours decreased.

### The relationship between sleep duration and hypertension

The results of the multivariable logistic regression analyses examined the relationship between sleep duration and hypertension were presented in table 4. Compared with 7-hour sleepers (reference group), the unadjusted analyses showed that short sleepers (<6 and 6 hours) had higher odds of hypertension (OR=1.47 and 1.19, respectively; p<0.05). After adjustment for age, sex, education level, occupation, marital status, annual household income, tea consumption, smoking status, alcohol intake, MET, sedentary leisure time, fresh fruit consumption, insomnia, sleep snoring, BMI, WC, DM, stroke and CHD, the increased odds of having hypertension was across those who reported ≥9 hours of sleep (OR=1.17, 95% CI 1.08 to 1.26; p<0.05). However, short sleep duration was no longer associated with hypertension.

There were similar results in men and women after adjustment for potential confounding factors. Long sleep duration was associated with elevated hypertension prevalence, with long sleepers (≥9 hours) having higher odds of hypertension in men (OR=1.16, 95% CI 1.04 to 1.30; p<0.05) and women (OR=1.19, 95% CI 1.08 to 1.32; p<0.05). The observed association between sleep duration and hypertension was slightly attenuated among men compared with women.

**Table 2**  Prevalence of hypertension in different gender groups among 55 687 adults in southwest China

| Variables | Male | | Female | | Overall | |
|---|---|---|---|---|---|---|
| | n (Prevalence %) | P value | n (Prevalence %) | P value | n (Prevalence %) | P value |
| Total | 6111 (28.67) | – | 7903 (22.99) | – | 14 014 (25.17) | <0.0001 |
| Age (years) | | | | | | |
| 30–39 | 417 (12.76) | <0.0001 | 331 (5.13) | <0.0001 | 748 (7.69) | <0.0001 |
| 40–49 | 905 (17.47) | | 1259 (12.82) | | 2164 (14.42) | |
| 50–59 | 1965 (28.45) | | 3243 (29.04) | | 5208 (28.81) | |
| 60–69 | 2115 (45.98) | | 2361 (42.75) | | 4476 (44.22) | |
| 70–79 | 709 (52.21) | | 709 (50.43) | | 1418 (51.30) | |
| Education level | | | | | | |
| No formal school | 847 (41.52) | <0.0001 | 2410 (36.78) | <0.0001 | 3257 (37.91) | <0.0001 |
| Primary school | 3481 (31.07) | | 3994 (24.12) | | 7475 (26.92) | |
| Middle school | 1235 (21.21) | | 1190 (13.13) | | 2425 (16.29) | |
| High school and above | 548 (24.37) | | 309 (14.08) | | 857 (19.28) | |
| Annual household income (¥) | | | | | | |
| <5000 | 2129 (37.15) | <0.0001 | 3000 (31.83) | <0.0001 | 5129 (33.84) | <0.0001 |
| 5000–9999 | 1852 (26.25) | | 2563 (20.29) | | 4415 (22.43) | |
| ≥10 000 | 2130 (24.97) | | 2340 (19.00) | | 4470 (21.44) | |
| Tea consumption | | | | | | |
| Never or almost never | 482 (33.04) | 0.0207 | 2335 (27.89) | <0.0001 | 2817 (28.66) | 0.0281 |
| Only occasionally | 1908 (28.19) | | 4396 (21.56) | | 6304 (23.22) | |
| Usually at least once a week | 3721 (28.43) | | 1172 (20.87) | | 4893 (26.16) | |
| Smoking status | | | | | | |
| Never | 726 (28.96) | <0.0001 | 5737 (20.95) | <0.0001 | 6463 (21.62) | <0.0001 |
| Ex-regular smoker | 857 (35.34) | | 520 (38.63) | | 1377 (36.52) | |
| Occasional smoker | 527 (25.41) | | 710 (30.14) | | 1237 (27.92) | |
| Smoked daily or almost every day | 4001 (27.96) | | 936 (28.54) | | 4937 (28.07) | |
| Alcohol intake | | | | | | |
| Never regular drinker | 663 (28.54) | <0.0001 | 4175 (24.88) | <0.0001 | 4838 (25.33) | <0.0001 |
| Ex-regular drinker | 572 (40.97) | | 257 (38.47) | | 829 (40.16) | |
| Occasional or seasonal drinker | 1646 (24.00) | | 2830 (19.14) | | 4476 (20.68) | |
| Usually at least once a week | 3230 (30.08) | | 641 (29.91) | | 3871 (30.05) | |
| Fresh fruit consumption | | | | | | |
| Daily | 549 (26.99) | <0.0001 | 819 (18.84) | <0.0001 | 1368 (21.44) | <0.0001 |
| 4–6 days/week | 482 (26.41) | | 755 (19.92) | | 1237 (22.03) | |
| 1–3 days/week | 2387 (26.61) | | 3249 (21.59) | | 5636 (23.46) | |
| Monthly | 1957 (30.08) | | 2364 (26.25) | | 4321 (27.86) | |
| Never/rarely | 736 (37.13) | | 716 (32.87) | | 1452 (34.90) | |
| Insomnia | | | | | | |
| No | 4650 (27.66) | <0.0001 | 5244 (21.01) | <0.0001 | 9894 (23.69) | <0.0001 |
| Yes | 1461 (32.42) | | 2659 (28.24) | | 4120 (29.59) | |
| Sleep snoring | | | | | | |
| Never | 2007 (25.39) | <0.0001 | 3391 (18.08) | <0.0001 | 5398 (20.25) | <0.0001 |
| Occasionally | 1481 (27.88) | | 1870 (24.94) | | 3351 (26.16) | |
| Usually | 2623 (32.39) | | 2642 (32.54) | | 5265 (32.47) | |
| BMI (kg/m$^2$) | | | | | | |
| <18.5 | 177 (19.80) | <0.0001 | 254 (16.13) | <0.0001 | 431 (17.46) | <0.0001 |

Continued

**Table 2** Continued

| Variables | Male | | Female | | Overall | |
|---|---|---|---|---|---|---|
| | n (Prevalence %) | P value | n (Prevalence %) | P value | n (Prevalence %) | P value |
| 18.5–23.9 | 3331 (24.51) | | 3170 (17.34) | | 6501 (20.40) | |
| 24–27.9 | 1999 (35.59) | | 3080 (27.68) | | 5079 (30.33) | |
| ≥28 | 604 (49.67) | | 1399 (41.28) | | 2003 (43.50) | |
| DM | | | | | | |
| No | 5726 (27.90) | <0.0001 | 7252 (21.94) | <0.0001 | 12 978 (24.23) | <0.0001 |
| Yes | 385 (48.67) | | 651 (49.17) | | 1036 (48.98) | |
| CHD | | | | | | |
| No | 6049 (28.54) | <0.0001 | 7839 (22.91) | <0.0001 | 13 888 (25.06) | <0.0001 |
| Yes | 62 (52.99) | | 64 (40.51) | | 126 (45.82) | |
| Stroke | | | | | | |
| No | 5999 (28.35) | <0.0001 | 7801 (22.79) | <0.0001 | 13 800 (24.91) | <0.0001 |
| Yes | 112 (73.20) | | 102 (73.91) | | 214 (73.54) | |
| Sleep duration (hours) | | | | | | |
| <6 | 1113 (33.68) | <0.0001 | 1933 (30.82) | <0.0001 | 3046 (31.81) | <0.0001 |
| 6 | 1108 (29.83) | | 1353 (25.66) | | 2461 (27.38) | |
| 7 | 1263 (27.67) | | 1423 (21.58) | | 2686 (24.07) | |
| 8 | 1803 (26.16) | | 2122 (19.47) | | 3925 (22.06) | |
| ≥9 | 824 (29.03) | | 1072 (20.09) | | 1896 (23.20) | |

'n (Prevalence %)' represents the number of hypertension and the prevalence. $\chi^2$ tests were conducted to test for differences in prevalence for unordered categorical variables and $\chi^2$ tests for trend were used to examine the trends for ordered categorical variables.
BMI, body mass index; CHD, coronary heart disease; DM, diabetes mellitus.

## The relationship between sleep duration and hypertension stratification by health-related factors

The results of covariate-adjusted analyses stratified by health-related factors examining relationship between sleep duration and hypertension were displayed in figure 1. Long sleep was associated with elevated hypertension prevalence for subgroups aged 45–59 and 60–79 years, though the relationship was nominally stronger among 45–59 individuals. However, this association was not found among subgroups aged 30–44 years. Among those usually at least once-a-week tea consumers, long sleep was associated with hypertension, and the relationship was not present across never/almost never or only occasionally tea consumers. Among those who were never regular alcohol drinkers, ex-regular or usually at least once-a-week alcohol drinkers, long sleep duration was associated with elevated hypertension prevalence. However, among occasional or seasonal alcohol drinkers, the relationship with long sleep was not present. Among different smoking status subgroups, long sleep was associated with hypertension among never smokers and daily or almost everyday smokers, and this association was not observed among ex-regular and occasional smokers. Long sleep was associated with elevated hypertension prevalence for non-regular fresh fruit consumers, while not among regular fresh fruit consumers. Long sleep was associated with elevated hypertension prevalence for insomnia subgroup, while not among non-insomnia subgroups.

## The dose–response relationship between sleep duration and hypertension by restricted cubic spline regression

The results of restricted cubic spline regression to calculate and visualise the relationship of sleep duration with hypertension were displayed in figure 2. The odds of hypertension was relatively flat until around 6.81 hours of sleep duration and then started to increase rapidly afterwards in subjects and a J-shaped pattern was observed (p for non-linearity=0.0023). Similar patterns were also observed in male subjects and more than 45-year-old subgroups. However, the non-linear tests were not significant for males and subjects more than 60 years old (p for non-linearity >0.05). There was a U-shaped relationship between sleep duration and hypertension in females (p for non-linearity=0.0081). A similar pattern was also observed in subjects aged 30–44 years, but the non-linear test was not significant (p for non-linearity=0.2481).

## The relationship between sleep duration and blood pressure

Age and gender-specific linear regression coefficients of sleep duration for SBP and DBP were presented in table 5. We estimated regression models separately for different age groups in two genders and then in the general population. After adjusting for covariates as

**Table 3** Prevalence of hypertension and distribution of patients with hypertension in different sleep duration groups among 55 687 adults in southwest China

| Variables | Prevalence of hypertension (%) | | | | | | n (%) | | | | | | |
|---|---|---|---|---|---|---|---|---|---|---|---|---|---|
| | <6 | 6 | 7 | 8 | ≥9 | P value | <6 | 6 | 7 | 8 | ≥9 | Overall | P value |
| Total | 31.81 | 27.38 | 24.07 | 22.06 | 23.20 | <0.0001 | 3046 (21.74) | 2461 (17.56) | 2686 (19.17) | 3925 (28.00) | 1896 (13.53) | 14 014 (100) | <0.0001 |
| **Gender** | | | | | | | | | | | | | |
| Female | 30.82 | 25.66 | 21.58 | 19.47 | 20.09 | | 1933 (24.46) | 1353 (17.12) | 1423 (18.01) | 2122 (26.85) | 1072 (13.56) | 7903 (100) | <0.0001 |
| Male | 33.68 | 29.83 | 27.67 | 26.16 | 29.03 | | 1113 (18.21) | 1108 (18.13) | 1263 (20.67) | 1803 (29.51) | 824 (13.48) | 6111 (100) | |
| P value | 0.0044 | <0.0001 | <0.0001 | <0.0001 | <0.0001 | | | | | | | | |
| **Age (years)** | | | | | | | | | | | | | |
| 30–39 | 7.45 | 7.52 | 8.49 | 7.48 | 7.58 | | 45 (6.02) | 75 (10.03) | 153 (20.45) | 310 (41.44) | 165 (22.06) | 748 (100) | <0.0001 |
| 40–49 | 15.38 | 13.18 | 13.85 | 14.39 | 15.60 | | 254 (11.74) | 275 (12.71) | 439 (20.29) | 795 (36.73) | 401 (18.53) | 2164 (100) | |
| 50–59 | 30.17 | 27.79 | 27.10 | 28.18 | 32.67 | | 1074 (20.62) | 911 (17.49) | 1034 (19.85) | 1483 (28.48) | 706 (13.56) | 5208 (100) | |
| 60–69 | 42.56 | 44.00 | 43.48 | 45.47 | 47.79 | | 1181 (26.39) | 902 (20.15) | 853 (19.06) | 1064 (23.77) | 476 (10.63) | 4476 (100) | |
| 70–79 | 49.90 | 51.83 | 50.49 | 52.20 | 54.81 | | 492 (34.70) | 298 (21.01) | 207 (14.60) | 273 (19.25) | 148 (10.44) | 1418 (100) | |
| P value | <0.0001 | <0.0001 | <0.0001 | <0.0001 | <0.0001 | | | | | | | | |
| **Tea consumption** | | | | | | | | | | | | | |
| Never or almost never | 35.05 | 29.14 | 27.32 | 24.63 | 27.15 | | 810 (28.75) | 447 (15.87) | 483 (17.15) | 663 (23.53) | 414 (14.70) | 2817 (100) | <0.0001 |
| Only occasionally | 30.38 | 26.23 | 22.10 | 19.79 | 19.95 | | 1465 (23.24) | 1177 (18.67) | 1190 (18.88) | 1721 (27.30) | 751 (11.91) | 6304 (100) | |
| Usually at least once a week | 31.56 | 28.22 | 25.29 | 24.07 | 25.34 | | 771 (15.76) | 837 (17.11) | 1013 (20.70) | 1541 (31.49) | 731 (14.94) | 4893 (100) | |
| P value | 0.0117 | 0.9427 | 0.7805 | 0.1173 | 0.9550 | | | | | | | | |
| **Smoking status** | | | | | | | | | | | | | |
| Never | 29.31 | 23.74 | 20.90 | 18.49 | 19.19 | | 1435 (22.20) | 1049 (16.23) | 1206 (18.66) | 1849 (28.61) | 924 (14.30) | 6463 (100) | <0.0001 |
| Ex-regular smoker | 39.16 | 38.07 | 34.10 | 34.74 | 37.37 | | 327 (23.75) | 257 (18.66) | 268 (19.46) | 346 (25.13) | 179 (13.00) | 1377 (100) | |
| Occasional smoker | 32.96 | 29.87 | 25.45 | 25.79 | 25.78 | | 294 (23.77) | 247 (19.97) | 227 (18.35) | 328 (26.51) | 141 (11.40) | 1237 (100) | |
| Smoked daily or almost every day | 33.53 | 29.60 | 26.55 | 25.38 | 27.93 | | 990 (20.05) | 908 (18.39) | 985 (19.95) | 1402 (28.40) | 652 (13.21) | 4937 (100) | |
| P value | 0.0001 | <0.0001 | <0.0001 | <0.0001 | <0.0001 | | | | | | | | |
| **Alcohol consumption** | | | | | | | | | | | | | |
| Never regular drinker | 33.14 | 26.73 | 24.31 | 22.22 | 22.05 | | 1190 (24.60) | 782 (16.16) | 869 (17.96) | 1340 (27.70) | 657 (13.58) | 4838 (100) | <0.0001 |
| Ex-regular drinker | 40.60 | 41.24 | 36.90 | 37.70 | 47.67 | | 188 (22.68) | 153 (18.46) | 162 (19.54) | 193 (23.28) | 133 (16.04) | 829 (100) | |
| Occasional or seasonal drinker | 27.73 | 23.87 | 19.71 | 17.43 | 17.90 | | 984 (21.98) | 825 (18.43) | 875 (19.55) | 1241 (27.73) | 551 (12.31) | 4476 (100) | |
| Usually at least once a week | 34.65 | 31.36 | 28.82 | 27.88 | 30.20 | | 684 (17.67) | 701 (18.11) | 780 (20.15) | 1151 (29.73) | 555 (14.34) | 3871 (100) | |
| P value | <0.0001 | <0.0001 | <0.0001 | <0.0001 | <0.0001 | | | | | | | | |
| **Fresh fruit consumption** | | | | | | | | | | | | | |

Continued

**Table 3** Continued

| Variables | Prevalence of hypertension (%) | | | | | | n (%) | | | | | | |
|---|---|---|---|---|---|---|---|---|---|---|---|---|---|
| | <6 | 6 | 7 | 8 | ≥9 | P value | <6 | 6 | 7 | 8 | ≥9 | Overall | P value |
| Daily | 28.13 | 23.41 | 22.97 | 18.49 | 19.61 | | 207 (15.13) | 202 (14.77) | 316 (23.10) | 410 (29.97) | 233 (17.03) | 1368 (100) | <0.0001 |
| 4–6 days/week | 26.26 | 26.41 | 21.34 | 19.63 | 20.88 | | 188 (15.20) | 215 (17.38) | 242 (19.56) | 379 (30.64) | 213 (17.22) | 1237 (100) | |
| 1–3 days/week | 29.34 | 25.56 | 23.18 | 20.97 | 21.43 | | 1082 (19.20) | 951 (16.87) | 1079 (19.14) | 1750 (31.06) | 774 (13.73) | 5636 (100) | |
| Monthly | 34.49 | 29.55 | 24.64 | 24.61 | 25.97 | | 1173 (27.15) | 854 (19.76) | 791 (18.31) | 1049 (24.27) | 454 (10.51) | 4321 (100) | |
| Never/rarely | 38.26 | 34.09 | 32.87 | 32.66 | 36.57 | | 396 (27.27) | 239 (16.46) | 258 (17.77) | 337 (23.21) | 222 (15.29) | 1452 (100) | |
| P value | <0.0001 | <0.0001 | <0.0001 | <0.0001 | <0.0001 | | | | | | | | <0.0001 |
| Insomnia | | | | | | | | | | | | | |
| No | 30.70 | 27.66 | 23.59 | 21.77 | 23.06 | | 634 (6.41) | 1621 (16.38) | 2248 (22.72) | 3617 (36.56) | 1774 (17.93) | 9894 (100) | <0.0001 |
| Yes | 32.11 | 26.85 | 26.89 | 26.24 | 25.42 | | 2412 (58.54) | 840 (20.39) | 438 (10.63) | 308 (7.48) | 122 (2.96) | 4120 (100) | |
| P value | 0.2228 | 0.4132 | 0.0040 | 0.0004 | 0.2347 | | | | | | | | 0.2347 |

'n (%) represents the number of hypertension and the percentages. The percentages in the table are row per cent. $\chi^2$ tests were conducted to test for differences in percentage for categorical variables. $\chi^2$ tests were conducted to test for differences in prevalence for unordered categorical variables and $\chi^2$ tests for trend were used to examine the trends for ordered categorical variables.

mentioned in table 4, sleep duration showed significant and positive associations with SBP and DBP both in males and females. Regression coefficients (SEs) of sleep duration and SBP were 0.474 (0.088), 0.417 (0.070) and 0.408 (0.055) in males, females and general population, respectively. Regression coefficients (SEs) of sleep duration and DBP were 0.203 (0.051), 0.102 (0.039) and 0.131 (0.031) in males, females and general population, respectively. There was a positive relationship between sleep duration and blood pressure, with 1-hour higher sleep duration associated with 0.474/0.203, 0.417/0.102 and 0.408/0.131 mm Hg higher SBP/DBP in males, females and general population, respectively. Age-specific linear regression coefficients of sleep duration for SBP and DBP were elevated with age. The association patterns were substantially similar for SBP in two genders. However, no significant associations were found between sleep duration and SBP in 30–44 years group among males. Only the association parameters for DBP were significant in males aged 45–59 years and in females aged 30–44 years.

## DISCUSSION

This study examined the relationship between sleep duration and hypertension among more than 50 000 rural adults in southwest China. Long sleep duration was associated with hypertension among men, women and the general population, which was consistent with several previous studies in different countries.[16–18 31] For example, the first study assessed the association between sleep duration and hypertension conducted by Gottlieb et al[17] and reported that compared with subjects sleeping 7 to less than 8 hours per night, the adjusted ORs (95% CIs) for hypertension of those sleeping 8 hours and more than 9 hours were 1.19 (1.04 to 1.37) and 1.30 (1.04 to 1.62), respectively. Long sleep duration of equal to or more than 8 hours per night was associated with hypertension in participants aged 40–100 years. The Nurses' Health Study[31] found long sleep duration (≥9 hours) was associated with elevated hypertension prevalence in a cross-sectional study. A systematic review and meta-analysis conducted by Guo et al[32] also found that long sleep duration was associated with hypertension (OR=1.11, 95% CI 1.04 to 1.18). Another meta-analysis conducted by Wang et al[33] indicated that the OR for long sleep duration and hypertension was 1.11 (95% CI 1.05 to 1.17).

However, numerous studies showed that compared with normal sleep duration, short sleep duration was more likely to be associated with hypertension than long sleep duration.[9–12 34 35] A cross-sectional study conducted among rural population in China showed that short sleep duration was associated with hypertension in women.[35] Another cross-sectional study in China including 43 655 participants found that short sleep duration of less than 6 hours was related to hypertension in both genders.[10] Results from the Kaiyuan study showed that short sleep duration (≤5 hours) was associated with an increased incidence of hypertension during the 3.98 years of follow-up

**Table 4** Multivariable-adjusted OR by logistic regression analyses of sleep duration associated with hypertension

| Population | Sleep duration (hours) | | | | |
| --- | --- | --- | --- | --- | --- |
| | <6 | 6 | 7 | 8 | ≥9 |
| Total,* n/N | 3046/9576 | 2461/8988 | 2686/11 159 | 3925/17 790 | 1896/8174 |
| Model 1 | **1.47 (1.38 to 1.56)** | **1.19 (1.12 to 1.27)** | 1 | 0.89 (0.84 to 0.94) | 0.95 (0.89 to 1.02) |
| Model 2 | 1.02 (0.96 to 1.09) | 0.99 (0.93 to 1.06) | 1 | 1.05 (0.99 to 1.11) | **1.19 (1.10 to 1.27)** |
| Model 3 | 1.02 (0.95 to 1.09) | 0.99 (0.93 to 1.06) | 1 | 1.05 (0.99 to 1.11) | **1.18 (1.10 to 1.27)** |
| Model 4 | 0.98 (0.91 to 1.05) | 0.98 (0.92 to 1.05) | 1 | 1.05 (0.99 to 1.12) | **1.16 (1.08 to 1.24)** |
| Model 5 | 0.96 (0.89 to 1.04) | 0.96 (0.90 to 1.03) | 1 | 1.06 (0.99 to 1.12) | **1.17 (1.08 to 1.26)** |
| Male,† n/N | 1113/3305 | 1108/3715 | 1263/4565 | 1803/6892 | 824/2838 |
| Model 1 | **1.33 (1.21 to 1.46)** | **1.11 (1.01 to 1.22)** | 1 | 0.93 (0.85 to 1.01) | 1.07 (0.96 to 1.19) |
| Model 2 | 0.97 (0.87 to 1.07) | 0.97 (0.88 to 1.07) | 1 | 1.03 (0.94 to 1.13) | **1.19 (1.07 to 1.33)** |
| Model 3 | 0.96 (0.86 to 1.06) | 0.97 (0.87 to 1.07) | 1 | 1.03 (0.95 to 1.13) | **1.19 (1.07 to 1.33)** |
| Model 4 | 0.92 (0.82 to 1.03) | 0.95 (0.86 to 1.05) | 1 | 1.04 (0.95 to 1.14) | **1.17 (1.05 to 1.31)** |
| Model 5 | 0.89 (0.79 to 1.00) | 0.93 (0.84 to 1.03) | 1 | 1.04 (0.95 to 1.14) | **1.16 (1.04 to 1.30)** |
| Female,† n/N | 1933/6271 | 1353/5273 | 1423/6594 | 2122/10 898 | 1072/5336 |
| Model 1 | **1.62 (1.50 to 1.75)** | **1.25 (1.15 to 1.37)** | 1 | 0.88 (0.82 to 0.95) | 0.91 (0.84 to 1.00) |
| Model 2 | 1.05 (0.97 to 1.15) | 1.01 (0.92 to 1.11) | 1 | 1.07 (0.98 to 1.15) | **1.20 (1.09 to 1.32)** |
| Model 3 | 1.05 (0.96 to 1.14) | 1.01 (0.92 to 1.11) | 1 | 1.07 (0.99 to 1.16) | **1.20 (1.09 to 1.32)** |
| Model 4 | 1.01 (0.91 to 1.11) | 1.00 (0.91 to 1.09) | 1 | 1.07 (0.98 to 1.16) | **1.17 (1.06 to 1.29)** |
| Model 5 | 1.00 (0.90 to 1.11) | 0.98 (0.89 to 1.07) | 1 | 1.07 (0.99 to 1.17) | **1.19 (1.08 to 1.32)** |

Data were analysed as ORs (95% CIs) for the 'presence of hypertension' relative to 'without hypertension' by logistic regression models using the 7-hour sleep duration as the reference group.

'n/N' represents the number of hypertension and the number of participants in each subgroup.

In bold are presented the statistically significant results (p value < 0.05).

*Model 1 was unadjusted; model 2 was adjusted for age and sex; model 3 was adjusted for model 2+education level, occupation, marital status and annual household income; model 4 was adjusted for model 3+tea consumption, smoking status, alcohol intake, metabolic equivalent (MET), sedentary leisure time, fresh fruit consumption, insomnia and sleep snoring; model 5 was adjusted for model 4+body mass index (BMI), waist circumference (WC), diabetes mellitus (DM), stroke and coronary heart disease (CHD).

†Model 1 was unadjusted; model 2 was adjusted for age; model 3 was adjusted for model 2+education level, occupation, marital status and annual household income; model 4 was adjusted for model 3+tea consumption, smoking status, alcohol intake, MET, sedentary leisure time, fresh fruit consumption, insomnia and sleep snoring; model 5 was adjusted for model 4+BMI, WC, DM, stroke and CHD.

period among women and those aged less than 60 years.[9] A prospective study conducted by Gangwisch *et al* reported that short sleep duration (≤5 hours) could significantly elevate the risk of hypertension (HR=2.10, 95% CI 1.58 to 2.79) in subjects between 32 and 59 years old.[11]

In this study, restricted cubic spline regression was conducted to visualise the relationship of sleep duration and hypertension. The odds of hypertension was relatively flat until around 6.81 hours of sleep duration and then started to increase rapidly afterwards and a J-shaped pattern was observed among adults. There was a U-shaped association between sleep duration and hypertension in females. However, inconsistent with many previous studies,[15–17 36] the association between short sleep duration and hypertension was not obvious in this study. There were plausible explanations for the different results of these studies. First, different definition of sleep duration groups and adjustment of different possible influencing factors would lead to differences in statistical analysis. Second, the discrepancy may be due to the difference of study population. Different demographic factors

and behavioural habits also would lead to the differences of the relationship between sleep duration and hypertension. Previous studies indicated that both insomnia[37] and snoring[38] could elevate the risk of hypertension, and our study adjusted these factors for multivariate analysis. In addition, the relationship of sleep duration and blood pressure was also explored in this study. After adjusting for potential confounders, sleep duration showed positive associations with blood pressure both in females and males. Regression coefficients of sleep duration and SBP/DBP were 0.474/0.203 in males and 0.417/0.102 in females. These results revealed that sleep duration was more closely associated with SBP than DBP.

The association between sleep duration and hypertension could reflect a variety of mechanisms. Several studies indicated that a potential pathophysiological mechanism supporting the association between short sleep duration and hypertension in humans may be sleep deprivation. Sleep deprivation has been reported as a cause of overactivity of the sympathetic nervous system, which may elevate blood pressure.[11 39] Sleep deprivation could disrupt

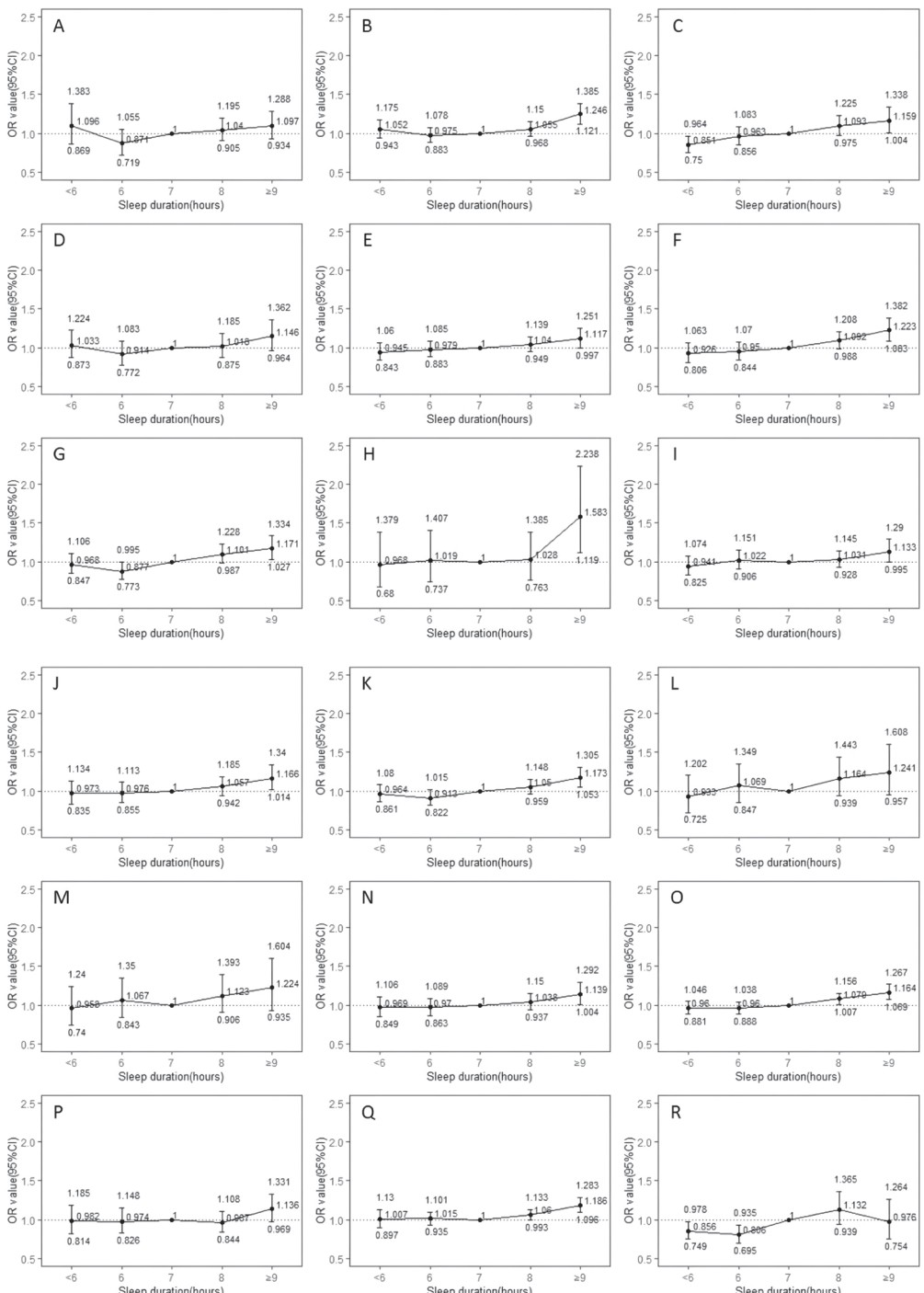

**Figure 1** Adjusted relationships between sleep duration and hypertension stratified by health-related factors. ORs in each group are presented as points with 95% CIs represented by extended lines. The horizontal reference dashed line represents an OR value of '1'. The full line represents the effects of different sleep durations on hypertension. (A) Subgroup aged 30–44 years. (B) Subgroup aged 45–59 years. (C) Subgroup aged 60–79 years. (D) Never or almost never tea consumers. (E) Only occasionally tea consumers. (F) Usually at least once-a-week tea consumers. (G) Never regular alcohol drinker. (H) Ex-regular alcohol drinker. (I) Occasional or seasonal alcohol drinker. (J) Usually at least once-a-week alcohol drinker. (K) Never smoker. (L) Ex-regular smoker. (M) Occasional smoker. (N) Daily or almost every day smoker. (O) Non-regular fresh fruit consumers. (P) Regular fresh fruit consumers. (Q) Insomnia subgroup. (R) Non-insomnia subgroup. Multivariable logistic model adjusted for age, sex, education level, occupation, marital status, income, tea consumption, smoking status, alcohol intake, MET, sedentary leisure time, fresh fruit consumption, insomnia, sleep snoring, BMI, WC, DM, stroke and CHD.

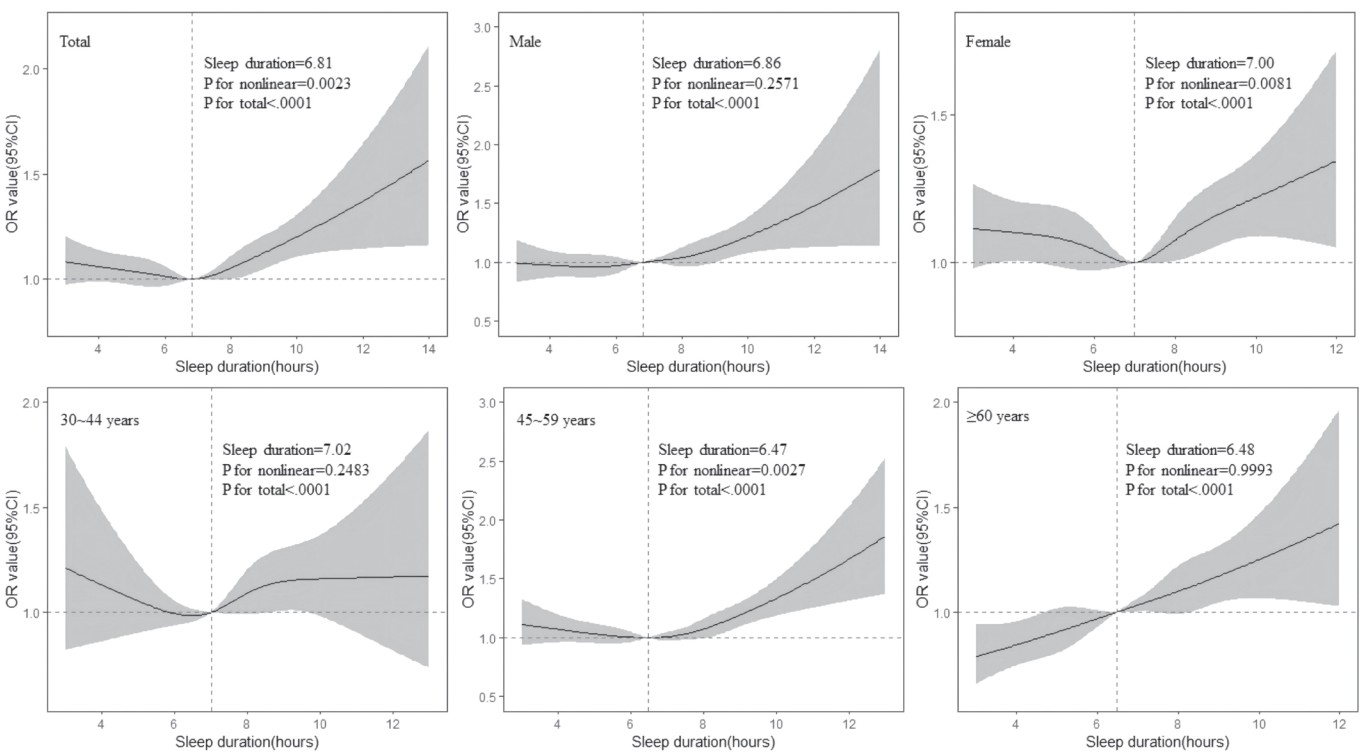

**Figure 2** Adjusted relationships between sleep duration and hypertension by using restricted cubic spline regression in different gender and age subgroups, adjusted by age, sex, education level, occupation, marital status, income, tea consumption, smoking status, alcohol intake, MET, sedentary leisure time, fresh fruit consumption, insomnia, sleep snoring, BMI, WC, DM, stroke and CHD. ORs are presented as full line with 95% CIs represented by grey area. The horizontal reference dashed line represents an OR value of '1'. The vertical dashed line represents the sleep duration where OR equals 1.

circadian rhythmicity, and contribute to overactivity of the renin-angiotensin aldosterone system, proinflammatory responses, endothelial dysfunction and renal impairment. Furthermore, sleep deprivation could increase appetite and suppress renal salt fluid excretion, leading to elevated caloric consumption, excessive salt intake and weight gain, which were proved to be risk factors of hypertension.[11 40] However, the mechanisms of long sleep duration with hypertension were even not known. Possible explanations may be as follows. First, some other risk factors might impact this association. Long sleepers may have less physical activities,[41] and be related to sleep disorders (sleep apnoea, etc) or poor sleep quality, which could lead to adverse healthy outcomes.[16] In previous studies, long sleep was associated with increased all-cause mortality, high risk of diabetes and other unhealthy outcomes.[41–43] Second, sleep duration may reflect sleep need, which is different in individuals due to genetic, behavioural and environmental factors.[17] Individuals with increased sleep needs may have a reduced physiological reserve, reducing their ability to survive serious illness.[41] Third, sleep habits may be a marker of health status rather than a cause for disease, so it is likely that long sleep duration is an epiphenomenon instead of an aetiological factor for hypertension.[18 44]

The relationship between sleep duration and hypertension may vary by health-related factors. This study made stratified analyses of sleep duration and hypertension stratified by age, tea consumption, alcohol intake, smoking status, fresh fruit consumption and insomnia. Long sleep was associated with elevated hypertension prevalence among participants aged 45–79 years. However, this association was not found among those aged 30–44 years. By restricted cubic spline regression, a J-shaped pattern was observed in subgroups aged 45–59 years. These findings were different from several previous studies. For example, a Korean study found that short sleep duration was associated with hypertension prevalence only in those aged less than 60 years.[45] A Spanish study demonstrated that sleep duration was not associated with hypertension in older adults.[46] A cross-sectional study conducted in northeast China observed an association between short sleep duration and hypertension in young adults aged 18–44 years, and the relationship was not found for participants aged 45–79 years subgroup.[19] Changes in sleep quality and quantity in different stage of life may be related to this age-dependent association.[47] In addition, healthy sleep was influenced by a lot of factors. Besides sleep duration, sleep disorders such as insomnia,[48 49] obstructive sleep apnoea[50] and other sleep quality problems[51–53] have also been shown to be risk factors for hypertension. Moreover, the number of younger subjects was larger than that of older subjects, so the lack of an association between long sleep duration

**Table 5** Regression coefficients (SEs) of sleep duration associated with SBP and DBP by gender and age groups

| Variables | SBP | | | | DBP | | | |
|---|---|---|---|---|---|---|---|---|
| | 30–44 years | 45–59 years | ≥60 years | Total | 30–44 years | 45–59 years | ≥60 years | Total |
| Overall* | | | | | | | | |
| Participants (n) | 18 787 | 24 013 | 12 887 | 55 687 | 18 787 | 24 013 | 12 887 | 55 687 |
| Regression coefficients (SEs) | 0.219 (0.073) | 0.424 (0.087) | 0.646 (0.131) | 0.408 (0.055) | 0.116 (0.050) | 0.153 (0.048) | 0.166 (0.069) | 0.131 (0.031) |
| P value | <0.001 | <0.001 | <0.001 | <0.001 | <0.05 | <0.01 | <0.05 | <0.001 |
| Male† | | | | | | | | |
| Participants (n) | 6364 | 8993 | 5958 | 21 315 | 6364 | 8993 | 5958 | 21 315 |
| Regression coefficients (SEs) | 0.213 (0.123) | 0.566 (0.134) | 0.554 (0.195) | 0.474 (0.088) | 0.072 (0.09) | 0.320 (0.079) | 0.194 (0.103) | 0.203 (0.051) |
| P value | >0.05 | <0.001 | <0.001 | <0.001 | >0.05 | <0.001 | >0.05 | <0.001 |
| Female† | | | | | | | | |
| Participants (n) | 12 423 | 15 020 | 6929 | 34 372 | 12 423 | 15 020 | 6929 | 34 372 |
| Regression coefficients (SEs) | 0.227 (0.091) | 0.378 (0.113) | 0.716 (0.178) | 0.417 (0.070) | 0.139 (0.061) | 0.070 (0.061) | 0.150 (0.092) | 0.102 (0.039) |
| P value | <0.05 | <0.001 | <0.001 | <0.001 | <0.05 | >0.05 | >0.05 | <0.01 |

*Age, sex, education level, occupation, marital status, income, tea consumption, smoking status, alcohol intake, metabolic equivalent (MET), sedentary leisure time, fresh fruit consumption, insomnia, sleep snoring, body mass index (BMI), waist circumference (WC), diabetes mellitus (DM), stroke and coronary heart disease (CHD) were included in models for adjustment.
†Age, education level, occupation, marital status, income, tea consumption, smoking status, alcohol intake, MET, sedentary leisure time, fresh fruit consumption, insomnia, sleep snoring, BMI, WC, DM, stroke and CHD were included in models for adjustment.
SBP, systolic blood pressure; DBP, diastolic blood pressure.

and hypertension in the younger subjects could be attribute to a lack of statistical power.

In this study, long sleep was associated with elevated hypertension prevalence for non-regular fresh fruit consumers, while not among subjects who ate fresh fruits regularly. Fresh fruit intake could reduce the risk and mortality of CVD. Results from previous prospective analyses of CKB showed that compared with participants who never or rarely consumed fresh fruits, those who ate fresh fruits daily had lower SBP (by 4 mm Hg, p<0.001). The adjusted HRs for daily consumers were 0.60 (95% CI 0.54 to 0.67) for cardiovascular death, and 0.66 (95% CI 0.58 to 0.75), 0.75 (95% CI 0.72 to 0.79) and 0.64 (95% CI 0.56 to 0.74) for incident major coronary events, ischaemic stroke and haemorrhagic stroke, respectively.[54] Another prospective study of CKB found that fresh fruit intake reduced all-cause mortality, CVD, cancer and chronic obstructive pulmonary disease mortality.[55] These results may explain the difference of relationship between long sleep duration and hypertension among non-regular and regular fresh fruit consumers. The association between long sleep duration and hypertension was more pronounced in individuals who had insomnia than non-insomnia subgroups. Two factors may explain the differential associations. First, insomnia has been proved to be a risk factor for hypertension.[37] Among insomnia participants, long sleep duration was more likely to be an unusual schedule, which could affect health outcomes in humans. Second, the sample size of non-insomnia subgroup was much higher than that of the insomnia group, which may lead to a statistical difference.

Our study had several strengths. This study was part of CKB, which was a large prospective cohort study in China. CKB had a series of standardised procedures and good quality control measures, so the data were relatively reliable. The sample size was large in this study, which could be a representative rural population in southwest China. Although this study revealed an association between long sleep duration and hypertension, several potential limitations of this study should be noted and need to be addressed in future work. First, our study was a cross-sectional study, so the directionality of the causation could not be established which did not allow for inferences regarding causation to be made. Further prospective cohort studies would be needed to confirm this association. Second, the data for sleep duration were self-reported without more detailed and objective measures of sleep, which might have information bias and were less accurate than actigraphy and polysomnography. Third, although we have attempted to adjust for various potential confounders, our results may still be subject to residual confounding from unmeasured or unknown influencing factors.

## CONCLUSIONS

In conclusion, this study identified that long sleep duration was significantly associated with hypertension and a J-shaped pattern was observed among adults in southwest China. However, this association was not found between short sleep duration and hypertension. It is important to sustain a proper amount of sleep duration for health. Further prospective studies are required to confirm these findings.

**Author affiliations**
[1]Department of Chronic and Non-communicable Disease Control and Prevention, Sichuan Center for Disease Control and Prevention, Chengdu, Sichuan, China
[2]Department of Epidemiology and Statistics, Chengdu Medical College, Chengdu, Sichuan, China
[3]Vanke School of Public Health, Tsinghua University, Beijing, China
[4]Global Health Research Center, Duke Kunshan University, Kunshan, Jiangsu, China
[5]Pengzhou Center for Disease Control and Prevention, Pengzhou, Sichuan, China
[6]Chinese Academy of Medical Sciences, Beijing, China
[7]Department of Epidemiology and Biostatistics, School of Public Health, Peking University, Beijing, China
[8]Center for Public Health and Epidemic Preparedness and Response, Peking University, Beijing, China
[9]Clinical Trial Service Unit (CTSU) and Epidemiological Studies Unit, Nuffield Department of Population Health, University of Oxford, Oxford, UK
[10]Sichuan Center for Disease Control and Prevention, Chengdu, Sichuan, China

**Acknowledgements** We thank the Chinese Center for Disease Control and Prevention, the Chinese Ministry of Health, the National Health and Family Planning Commission of China and 10 provincial/regional health administrative departments. The most important acknowledgement is to the participants in the study and the members of the survey teams in each of the 10 regional centres, as well as to the project development and management teams based in Beijing, Oxford and the 10 regional centres.

**Contributors** XW and GL conceived and designed the paper. LL and ZC designed and supervised the conduct of the CKB study, obtained funding and, together with YG and PP, acquired the data. XChang, XChen and JSJ analysed the data. XChang, XChen*, QS and NZ drafted the manuscript. XChang, XChen, JSJ, GL, XChen*, QS, NZ and XW contributed to the interpretation of the results and critical revision of the manuscript. All authors contributed to and approved the final manuscript. XW is the study guarantor.

**Funding** This work was supported by grants ('Tianfu ten thousand talents plan' fund in 2018) from the Sichuan Provincial Leading Group Office for Talent Work, grants from the National Key Research and Development Program of China (2016YFC0900500, 2016YFC0900501, 2016YFC0900504), grants from the Kadoorie Charitable Foundation in Hong Kong and grants from the Wellcome Trust (212946/Z/18/Z, 202922/Z/16/Z, 104085/Z/14/Z, 088158/Z/09/Z) in the UK.

**Disclaimer** The funders had no role in the study design, data collection, data analysis and interpretation, writing of the report or the decision to submit the article for publication.

**Competing interests** None declared.

**Patient and public involvement** Patients and/or the public were not involved in the design, or conduct, or reporting, or dissemination plans of this research.

**Patient consent for publication** Not required.

**Ethics approval** This study involves human participants and was approved by the Ethical Review Committee of the Chinese Center for Disease Control and Prevention (Beijing, China) (ethics approval ID number: 005/2004) and the Oxford Tropical Research Ethics Committee, University of Oxford (UK) (ethics approval ID number: OXTREC 025-04). Participants gave informed consent to participate in the study before taking part.

**Provenance and peer review** Not commissioned; externally peer reviewed.

**Data availability statement** Data are available upon reasonable request. The data set supporting the conclusions of this article is available from the study website (http://www.ckbiobank.org), along with the access policy and procedures.

**ORCID iDs**
Xiaoyu Chang http://orcid.org/0000-0001-8361-3977
John S Ji http://orcid.org/0000-0002-5002-118X

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
