## [Reviewer comments · BMJ Open]

ARTICLE DETAILS

TITLE (PROVISIONAL)	The association between sleep duration and hypertension in southwest China: a population-based cross-sectional study
AUTHORS	Chang, Xiaoyu; Chen, Xiaofang; Ji, John; Luo, Guojin; Chen, Xiaofang; Sun, Qiang; Zhang, Ningmei; Guo, Yu; Pei, Pei; Li, Liming; Chen, Zhengming; Wu, Xianping

VERSION 1 – REVIEW

REVIEWER	Chen, Weihong Huazhong University of Science and Technology
REVIEW RETURNED	31-May-2021

GENERAL COMMENTS	The authors have analyzed the association between sleep duration and hypertension in southwest China with a cross-sectional study. Overall, the sample size is large and the age group is more, which could provide more information. But some comments need to be addressed. (1) The sleep duration is 6, 7, 8, or 6-7, 7-8, 8-9? Please clarify (2) The dose response relationship with cure spline is needed in the result. (3) Please add the relationship of sleep duration with sbp and dbp (4) The English language is need to improved: a) The “risk” of hypertension should be avoided, as it is a cross-sectional study. b) Long sleep duration of longer sleep duration ? c)
---

REVIEWER	Nieminen, Pentti University of Oulu
REVIEW RETURNED	02-Jun-2021

GENERAL COMMENTS	This review is primarily a statistical one, with recommendations and specific major and minor points. 1) The quality of statistical reporting and data presentation was acceptable. I score 6 in a scale from 0 (poor) to 10 (very high). However, the authors could improve it. 2) Page 8, Statistical analysis sub-section: There are insufficient details in the statistical methods, and it needs to be more detail. For example: - Line 16: Replace the symbol of mean value with correct symbol and define the abbreviations. - Line 18: The expression “composition ratio or rate” is strange. Do you mean “Frequency and percentage distributions were reported for categorical variables”?
---

	- Lines 18-22: Please name the continuous and categorical variables (covariates). In addition, clarify that you are estimating the statistical significance of their associations with sleep duration. You should also estimate their associations with your main outcome, i.e. hypertension. - Line 26: Name the potential confounding factors analysed. - Have you examined normality of the continuous covariates? 3) Your main outcome variable is hypertension (no vs yes). Help your readers and include a table where frequency and percentage distributions of hypertension are presented by sleep duration and the main covariates used for the adjustment in the multivariable regression modelling. This data helps your readers to interpret why the relationship between sleep duration and hypertension changed from the unadjusted analysis. 4) Table 2: The adjustment for age and sex changed the effect of sleep duration on the outcome. Were age and sex confounding factors? Table 1 clearly shows that age is associated with sleep duration. In the Figure 1 you report ORs stratified by age groups. In Table 2, you report ORs stratified by age and sex. However, in models 2 for males and females, the only adjusting factor is age. Further adjustments with other covariates do not seem to change the findings. How is age related with hypertension? Readers would like to see the relationship between age and hypertension. Is it age that explains the unadjusted correlation between sleep duration and hypertension? I think you should clarify this more clearly using your data. Please see my previous comment about adding a table where the distribution of hypertension is reported by sleep duration, age and some other (but not all) potential confounding factors. 5) Table 1: The title is not correct. You report distributions of sleep duration by categorical basic characteristics and descriptive statistics of continuous covariates in the sleep duration groups. In addition, divide the table more clearly to two parts. In the first part add "n (%)" to column headings and report the p-value of chi-square test. In the second part add "Mean (SD)" to column headings.
--	---

VERSION 1 – AUTHOR RESPONSE

Reviewer: 1

Dr. Weihong Chen, Huazhong University of Science and Technology

Comments to the Author:

The authors have analyzed the association between sleep duration and hypertension in southwest China with a cross-sectional study. Overall, the sample size is large and the age group is more, which could provide more information. But some comments need to be addressed.

(1) The sleep duration is 6, 7, 8, or 6-7, 7-8, 8-9? Please clarify

Responds: At the baseline survey, participants were asked to report the number of hours they slept a day during the last 12 months. Sleep duration was assessed by a self-reported questionnaire with the following question: "On average, how many hours do you typically sleep per day (including daytime naps)?". Respondents could report in only 1-hour increments. In this study, sleep duration was categorized as five groups of <6, 6, 7, 8 and ≥ 9 hours. Therefore, the sleep duration is 6, 7, 8 hours, not 6-7, 7-8, 8-9 hours. We have revised the description of sleep duration in the part of "subject and methods".

(2) The dose response relationship with cure spline is needed in the result.

Responds: Done. Restricted cubic splines regression was used to visualize the relationship of sleep duration with hypertension. Results of restricted cubic splines regression among different gender and age subgroups were displayed in Figure 2.

(3) Please add the relationship of sleep duration with sbp and dbp

Responds: Done. We used multiple linear regression model to analyse the relationship between sleep duration and blood pressure. Results of the association for sleep duration with sbp and dbp were displayed in Table 4. Age and gender-specific linear regression coefficients of sleep duration for SBP and DBP were calculated by multiple linear regression model.

(4) The English language is need to improved:

a) The “risk” of hypertension should be avoided, as it is a cross-sectional study.

b) Long sleep duration of longer sleep duration ?

c)

Responds:

a) Yes, have changed throughout the manuscript.

b) It means long sleep duration. We have changed throughout the manuscript.

Reviewer: 2

Dr. Pentti Nieminen, University of Oulu

Comments to the Author:

This review is primarily a statistical one, with recommendations and specific major and minor points.

1) The quality of statistical reporting and data presentation was acceptable. I score 6 in a scale from 0 (poor) to 10 (very high). However, the authors could improve it.

Responds: According to the comments and suggestions of reviewers and editor, we have revised our manuscript.

2) Page 8, Statistical analysis sub-section: There are insufficient details in the statistical methods, and it needs to be more detail. For example:

- Line 16: Replace the symbol of mean value with correct symbol and define the abbreviations.

Responds: Done. Have changed at the beginning of statistical analysis sub-section.

- Line 18: The expression “composition ratio or rate” is strange. Do you mean “Frequency and percentage distributions were reported for categorical variables”?

Responds: Yes. Have changed throughout the manuscript.

- Lines 18-22: Please name the continuous and categorical variables (covariates). In addition, clarify that you are estimating the statistical significance of their associations with sleep duration. You should also estimate their associations with your main outcome, i.e. hypertension.

Responds: Done. Have changed throughout the manuscript.

In statistical analysis sub-section, we named the continuous and categorical variables (covariates) and clarified that the statistical significance of their associations with sleep duration was estimated. Results were presented in Table 1.

We also estimated associations of the continuous and categorical variables with hypertension. We added two tables to display the prevalence of hypertension in different covariates subgroups. Chi-square tests and the Cochran-Armitage tests for trend were used to examine the relationships between covariates and hypertension. Results were presented in Table 2 and Table 3.

- Line 26: Name the potential confounding factors analysed.

Responds: Done. Have changed throughout the manuscript.

- Have you examined normality of the continuous covariates?

Responds: Yes, we examined normality of the continuous covariates.

3) Your main outcome variable is hypertension (no vs yes). Help your readers and include a table where frequency and percentage distributions of hypertension are presented by sleep duration and the main covariates used for the adjustment in the multivariable regression modelling. This data helps

your readers to interpret why the relationship between sleep duration and hypertension changed from the unadjusted analysis.

Responds: Done. We added two tables to display the prevalence of hypertension and percentage distributions of hypertension in different sleep duration groups and the main covariates used for the adjustment in the multivariable regression modelling subgroups. Results were presented in Table 2 and Table 3.

4) Table 2: The adjustment for age and sex changed the effect of sleep duration on the outcome. Were age and sex confounding factors? Table 1 clearly shows that age is associated with sleep duration. In the Figure 1 you report ORs stratified by age groups. In Table 2, you report ORs stratified by age and sex. However, in models 2 for males and females, the only adjusting factor is age. Further adjustments with other covariates do not seem to change the findings. How is age related with hypertension? Readers would like to see the relationship between age and hypertension. Is it age that explains the unadjusted correlation between sleep duration and hypertension? I think you should clarify this more clearly using your data. Please see my previous comment about adding a table where the distribution of hypertension is reported by sleep duration, age and some other (but not all) potential confounding factors.

Responds: Done. We added a table where the distribution of hypertension is reported by sleep duration, age and some other potential confounding factors. We also analyzed the relationships between sleep duration and hypertension, SBP and DBP by sex and age groups by restricted cubic splines and multiple linear regression model.

5) Table 1: The title is not correct. You report distributions of sleep duration by categorical basic characteristics and descriptive statistics of continuous covariates in the sleep duration groups. In addition, divide the table more clearly to two parts. In the first part add "n (%)" to column headings and report the p-value of chi-square test. In the second part add "Mean (SD)" to column headings.

Responds: Done. We have revised the title of Table 1. We added "n (%)" and "Mean (SD)" to column headings of Table 1. Have changed throughout the manuscript.

VERSION 2 – REVIEW

REVIEWER	Nieminen, Pentti University of Oulu
REVIEW RETURNED	24-Sep-2021

GENERAL COMMENTS	The authors addressed the main concerns from the reviews, the revised version of the manuscript appears to be good. I have some minor comments:  1. Page 8, lines 43-45: You responded to my comment about the assumptions for the analysis of variance that you have examined them. Please say it also to your readers. Clarify that the data satisfactorily fulfil the underlying assumptions and preconditions of the main analysis methods. 2. Page 9, line 4: Please give references to restricted cubic splines regression. 3. Table 5: Please do not denote p-values with asterisks, and actual p-values were not reported. Actual p-values should be reported, without false precision, whenever feasible. Providing the actual p-values prevents problems of interpretation related to p-values close to 0.05. Very small p-values do not need exact representation and $p < 0.001$ is usually sufficient. 4. Table 5: The regression coefficients for categorized SBP, DBP and Total are difficult to interpret. Have you estimated regression
--

	models separately for three blood pressure groups and then in the total sample? Consider clarifying their interpretation. 5. Tables 4 and 5: Consider providing the total number of participants or number of participants in each subgroup.
--	--

VERSION 2 – AUTHOR RESPONSE

Reviewer: 2

Dr. Pentti Nieminen, University of Oulu

Comments to the Author:

The authors addressed the main concerns from the reviews, the revised version of the manuscript appears to be good.

I have some minor comments:

1. Page 8, lines 43-45: You responded to my comment about the assumptions for the analysis of variance that you have examined them. Please say it also to your readers. Clarify that the data satisfactorily fulfil the underlying assumptions and preconditions of the main analysis methods.

Responds: Done. We added the description that we have examined the assumptions for the analysis of variance in the statistical analysis sub-section. Have changed throughout the manuscript.

2. Page 9, line 4: Please give references to restricted cubic splines regression.

Responds: Done. Have changed throughout the manuscript.

3. Table 5: Please do not denote p-values with asterisks, and actual p-values were not reported. Actual p-values should be reported, without false precision, whenever feasible. Providing the actual p-values prevents problems of interpretation related to p-values close to 0.05. Very small p-values do not need exact representation and $p < 0.001$ is usually sufficient.

Responds: Done. We deleted asterisks and added columns of p values in Table 5. Besides, We also found that we denoted p-values with asterisks in Table 4. Since p values were described in the description section of Table 4, we deleted asterisks in Table 4. Have changed throughout the manuscript.

4. Table 5: The regression coefficients for categorized SBP, DBP and Total are difficult to interpret. Have you estimated regression models separately for three blood pressure groups and then in the total sample? Consider clarifying their interpretation.

Responds: We used multiple linear regression model to analyse the relationship between sleep duration and blood pressure in different gender and age groups. We estimated regression models separately for different gender and age groups, and then in the total sample. Age and gender-specific

linear regression coefficients of sleep duration for SBP and DBP were calculated. We added description of regression coefficients for sleep duration and blood pressure in the sub-section of Table 5. Have changed throughout the manuscript.

5. Tables 4 and 5: Consider providing the total number of participants or number of participants in each subgroup.

Responds: Done. We have revised Table 4 and Table 5. Have changed throughout the manuscript.

Reviewer: 2

Competing interests of Reviewer: I have no competing interests

VERSION 3 – REVIEW

REVIEWER	Nieminen, Pentti University of Oulu
REVIEW RETURNED	22-Feb-2022
GENERAL COMMENTS	I thank the authors for careful consideration of the comments and have no further suggestions at this point.